# Leveraging Dark Knowledge for Intrinsic Multimodal Out-of-Distribution Detection

## Abstract

Out-of-distribution (OOD) detection is crucial for the safe deployment of deep neural models in applications such as autonomous driving. With the emerging multimodal nature of modern applications, recent attention has shifted toward OOD detection in multimodal settings. However, current multimodal OOD detection methods fail to fully exploit the synergy among modalities: they treat all modalities equally, disregarding their varying detection performance, and they are unable to capture the diverse uncertainty information encoded at the logit level. In this paper, we propose to exploit the *dark knowledge* within unimodal experts as the key to revealing their synergy. To this end, we introduce a self multimodal OOD distillation framework, which leverages logits as uncertainty-aware soft targets to train a holistic model that operates in the joint embedding space of all modalities. Specifically, the proposed framework accounts for the negative effects of underperforming modalities and effectively fuses both the rich feature-level knowledge and the logit-level knowledge of modalities. As a result, our method improves the performance of current state-of-the-art multimodal OOD detection methods, achieving gains of up to 30% across diverse OOD detection benchmarks, spanning two tasks and five multimodal OOD datasets.

## 1 Introduction

Out-of-distribution (OOD) detection aims to reliably identify samples that differ from the training distribution, which is a crucial requirement for deploying deep neural networks in safety-critical domains such as autonomous driving (Liu et al., 2023b) and healthcare (Fink et al., 2020). Consequently, a significant body of research has been dedicated to OOD detection (Yang et al., 2024), where the most standard and fundamental approach is to derive a confidence score from a deep neural classifier (Hendrycks & Gimpel, 2017; Liu et al., 2020; Hendrycks et al., 2022). Many other approaches have also been proposed, ranging from distance-based methods (Sun et al., 2022; Lee et al., 2018) to retraining strategies that enhance ID–OOD separability (Hendrycks et al., 2019; Du et al., 2022). However, they have predominantly focused on single-modality inputs, such as images or videos, despite the inherently multimodal nature of real-world applications (Li et al., 2025).

More recently, Dong et al. (2024) introduced the first multimodal OOD benchmark, enabling methods to leverage the complementary nature of various modalities for OOD detection. They observed that distributional shift manifests as increased diversity in predictions across modalities and proposed an algorithm to amplify the OOD discrepancy during training. Since then, several works have been proposed (Li et al., 2025; Liu et al., 2025) for multimodal OOD detection. However, these methods are unable to capture the full synergy between modalities due to several limiting factors. For example, they employ complex retraining strategies to enhance the embedding space for better OOD separation, including contrastive learning (Li et al., 2025), maximizing disagreement between models on OOD samples (Dong et al., 2024), or improving supervision from unknown data via outlier synthesis (Liu et al., 2025), while paying limited attention to the intrinsic OOD detection capability of unimodal models. A final classifier is then employed by fusing the enhanced embeddings from unimodal models, incorporating one-hot targets. However, they overlook diverse uncertainty knowledge contained at the logit level of each unimodal expert, which can be incorporated to strengthen OOD detection. Particularly, a key challenge underlying OOD overconfidence is that *ground-truth* uncertainty labels for training samples are unavailable (Lakshminarayanan et al., 2017). Instead, only one-hot class labels are available, which cause the model to treat all class

samples uniformly, despite their varying degrees of uncertainty, potentially leading to overconfident predictions (Yang & Xu, 2025). On the other hand, we found that, as illustrated in Fig. 1, some modalities (e.g., video) can achieve stronger detection performance than others (e.g., optical flow)—a fact largely overlooked in previous research, where all modalities are treated equally without accounting for their varying detection capabilities.

To address these challenges and fully exploit the synergy across modalities for OOD detection, we propose a self multimodal OOD distillation framework, which trains a holistic model whose guidance is self-provided by unimodal experts serving as teachers. Traditionally, the hidden information within the teacher has been referred to as *dark knowledge* (Hinton et al., 2015). Accordingly, our framework leverages the uncertainty-aware dark knowledge encoded at the logit level of unimodal experts to strengthen multimodal OOD detection performance. Given that ground-truth uncertainty is unavailable, we first estimate predictive uncertainty by approximating it through the collective behavior of all models, while accounting for underperforming modalities. We then leverage this estimated uncertainty as soft targets to train a joint classifier in the embedding space shared across all modalities, thereby reducing OOD overconfidence and accounting for disparities among modalities. This joint classifier can be conceptually regarded as the student, guided by the same ensemble of unimodal teachers, who also collectively share the embedding space.

In this way, our method exploits both the diverse uncertainty knowledge encoded at the logit level and the representational knowledge at the feature level of unimodal experts, thereby revealing their ultimate synergy. As part of this work, we also provide a comprehensive analysis of how the uncertainty-aware dark knowledge in our framework contributes to improving multimodal OOD detection performance, which cannot be achieved by naively combining ensembles or merely fusing features across modalities.

This principled and effective framework of ours allows us to integrate it as a plug-in to mine the intrinsic OOD detection capabilities of a given multimodal model set, thereby alleviating the burden of retraining large modality-specific models. A simple integration of our framework into a vanilla baseline yields substantial improvements in OOD performance (Fig. 1), achieving comparable or even superior results to current retraining-based state-of-the-art multimodal OOD detection methods. Moreover, being agnostic to these methods, it further improves their performance with gains up to 30% across diverse OOD detection benchmarks, spanning two tasks and five OOD datasets.

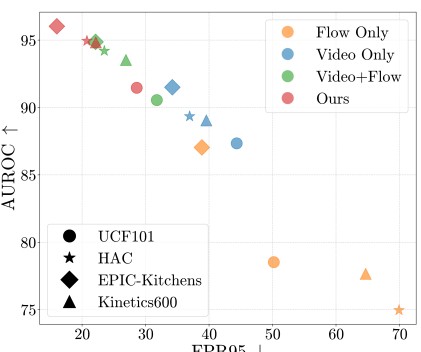

Figure 1: OOD detection performance on the HMDB51 dataset across different modalities. The detection performance of *Flow* is substantially lower than that of *Video*. Multimodal OOD detection (*Video+Flow*, obtained using one of the SOTA methods (Dong et al., 2024)) improves over unimodal OOD, while our method more effectively harnesses the synergy among modalities.

## 2 RELATED WORK

**Out-of-Distribution (OOD) Detection.** A primary approach to OOD detection is to leverage confidence scores derived from deep neural classifiers, such as the maximum softmax probability (MSP) (Hendrycks & Gimpel, 2017), Energy (Liu et al., 2020)—which also mirrors the class-conditional probability—Generalized Entropy (GEN) (Liu et al., 2023a), and the maximum of logit (MaxLogit) (Hendrycks et al., 2022). In contrast to these classifier-based scores, kNN (Sun et al., 2022) and Mahalanobis (Lee et al., 2018) rely on distance metrics in feature space for OOD detection, while Virtual Logit Matching (VIM) (Wang et al., 2022) integrates information from both feature and logit spaces to define the OOD score. Other approaches, such as ReAct (Sun et al., 2021) and ASH (Djurisic et al., 2022), enhance existing scoring functions by modifying input activations. In contrast to the above post-hoc methods, retraining-based approaches (Hendrycks et al., 2019; Du et al., 2022) aim to enhance representations for better ID–OOD separability. However, all these methods predominantly focus on single-modality inputs, without accounting for the multimodal nature of recent applications.

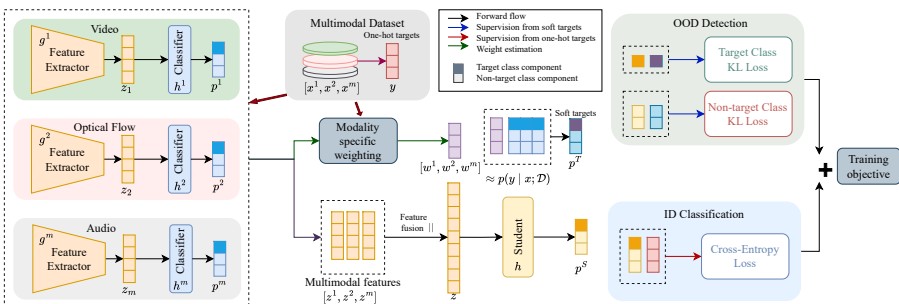

Figure 2: An overview of the proposed self multimodal OOD distillation framework.

**Multimodal OOD Detection.** With the emergence of large vision–language models (Radford et al., 2021), the task of *multimodal OOD detection* was initially associated with approaches that exploited semantic information from text labels (Ming et al., 2022; Wang et al., 2023b), although their scope remained restricted to image-only benchmarks. To address this, Dong et al. (2024) introduced the first multimodal OOD benchmark incorporating multiple modality combinations (i.e., video, audio, and optical flow), and observed that OOD samples tend to exhibit higher discrepancy among unimodal prediction distributions. They proposed the Agree-to-Disagree algorithm to amplify this discrepancy during training and further introduced multimodal outlier synthesis (NP-Mix) to obtain a more discriminative embedding space for OOD detection. More recently, DPU (Li et al., 2025) employed a contrastive learning strategy to account for intra-class variability in multimodal OOD detection, while Feature Mixing (Liu et al., 2025) introduced yet another synthetic outlier–based approach for training. However, these methods are unable to reveal the full synergy between modalities, in that they treat all modalities equally, without considering their varying detection performance. In addition, they are unable to capture holistic uncertainty across modalities, leaving the rich uncertainty-aware information within the logit layers of unimodal experts untapped. Moreover, they rely on complex retraining approaches, especially with a focus on providing a more discriminative embedding space for OOD detection, while little attention has been given to harnessing the intrinsic OOD detection capability of the ensemble itself.

**Logit Distillation.** Logit distillation, which transfers the *dark knowledge* within logits of a large model to a smaller one, was initially proposed for model compression (Hinton et al., 2015), but has since been shown to also improve model accuracy. For example, retraining with self-distillation, has been found to further improve accuracy (Zhang et al., 2019). Zhao et al. (2022) sought to shed light on dark knowledge by decoupling it into two components: binary knowledge related to the target class and relational knowledge pertaining to the non-target classes. More closely related to our work, Yang & Xu (2025) employed self-knowledge distillation for OOD detection to mitigate the negative effect from atypical samples during training. However, their method is limited to single modalities, whereas we leverage it to harness the synergy among multiple modalities. Logit distillation has also been explored in multimodal applications for cross-modal adaptation (Radevski et al., 2023), but not to OOD detection. To the best of our knowledge, this is the first work that methodically leverages and analyzes the role of dark knowledge in improving multimodal OOD detection.

## 3 METHODOLOGY

In this section, we present our self multimodal OOD distillation framework. We begin with a formal introduction to the multimodal OOD detection problem, followed by an overview of the framework and a detailed description of the method. We also provide further insights into uncertainty-aware dark knowledge within our framework at the end of this section.

### 3.1 PROBLEM DEFINITION

Let $\mathcal{D}_{\text{in}} = \{(\boldsymbol{x}_i, y_i)\}_{i=1}^n$ be our training set, where each $\boldsymbol{x}_i \in \mathcal{X}$ is an input sample and $y_i \in \mathcal{Y} = \{1, 2, \ldots, C\}$ is the corresponding class label. In a multimodal setting each training sample $\boldsymbol{x}_i$ is comprised of $M$ modalities, denoted as $\boldsymbol{x}_i = \{\boldsymbol{x}_i^m \mid m = 1, \ldots, M\}$. Let $M$ be a set of neural networks trained on each unimodality (e.g., video), $f^m : \mathcal{X}^m \to \mathbb{R}^C$. The goal of multimodal OOD

detection is to effectively combine information from all modalities in order to correctly identify samples with semantic shifts. Each model $f^m$ comprises a feature extractor $g^m(\cdot)$, which extracts an embedding $\boldsymbol{z}^m$ for its corresponding modality $m$ and the usual practice is to employ a final classifier $h(\cdot)$, which takes the combined embeddings from all modalities as input and outputs a prediction probability $\boldsymbol{p}^{\mathrm{S}} = \delta(h([\,g^1(\boldsymbol{x}^1)\,||\,\ldots\,||\,g^M(\boldsymbol{x}^M)\,])) = \delta(h([\,\boldsymbol{z}^1\,||\,\ldots\,||\,\boldsymbol{z}^M\,])) = \delta(h(\boldsymbol{z}))$, where $||$ denotes concatenation and $\delta(\cdot)$ denotes the softmax function.

When deploying in the real world, $h$ should not only accurately classify known samples from the in-distribution (ID), but also detect samples that exhibit semantic shifts compared to ID samples and do not belong to any class in $\mathcal{Y}$, i.e., out-of-distribution (OOD) samples. This aims to define a separate scoring function $S : \mathcal{X} \to \mathbb{R}$, which assigns a high score $S(\boldsymbol{x})$ for samples $\boldsymbol{x} \sim \mathcal{D}_{\mathrm{in}}$ and a low score otherwise. A threshold $\eta$ is then used to classify each sample $\boldsymbol{x} \in \mathcal{X}$ as ID or OOD:

$$\mathrm{Decision}(\boldsymbol{x}) = \begin{cases} \mathrm{ID}, & \text{if } S(\boldsymbol{x}) \geq \eta, \\ \mathrm{OOD}, & \text{if } S(\boldsymbol{x}) < \eta. \end{cases}$$

### 3.2 MOTIVATION AND METHOD OVERVIEW

In this section, we present the motivation behind our approach and provide a brief overview of our framework illustrated in Fig. 2. A common approach to combining complementary knowledge from different modalities is to train a joint classifier ($h$) in the combined embedding space of modalities, which yields improvements in OOD detection over unimodal models (see 'Video+Flow' in Fig. 1). However, this alone is unable to fully leverage the synergy between modalities, as it does not properly account for underperforming modalities, and it fails to fully exploit the uncertainty-related knowledge encoded in modality-specific models. As shown in Fig. 1, some modalities (e.g., optical flow) exhibit weaker OOD detection performance than others (e.g., video). Specifically, this disparity across modalities–whose weakness partly arises from OOD overconfidence– is difficult to address solely at the feature level; therefore, we turn to the logit level for complementary guidance. In particular, a key challenge underlying OOD overconfidence is that *ground-truth* uncertainty labels for training samples are not available. Therefore, we first model the predictive uncertainty by treating the collective behavior of all models as an approximation to the true uncertainty. We then employ logit distillation to leverage this estimated uncertainty as soft targets to train the joint classifier ($h$). In this framework, $h$ acts as the student, and its guidance is self-provided by the set of unimodal teachers that collectively share the embedding space. We also employ decoupled KL divergence (Zhao et al., 2022) for logit distillation, separating it into target and non-target class components, which allows better weight adjustment across tasks of varying difficulty (e.g., near-OOD vs. far-OOD). Ultimately, our framework not only fuses rich feature-space knowledge but also leverages the uncertainty-aware knowledge residing in the logit space of unimodal teachers in a weighted manner, thus revealing the synergy among them. Building on these insights, we present a detailed and formal description of our framework in the next section.

### 3.3 SELF MULTIMODAL OOD DISTILLATION FRAMEWORK

To improve the multimodal OOD detetcion we focus on both feature-space and logit-space knowledge of multimodal models. We begin with the joint classifier $h$ that operates on the combined embeddings space of unimodal models $\{f^m\}$. In order to model the holistic predictive capability of all modalities, we further include a separate classifier $h^m(\cdot)$ for each modality $m$ in order to obtain predictions from each modality individually. The conditional distribution over classes from the $m$-th modality is then given by $p(y \mid \boldsymbol{x}^m, f^m) := \delta(h^m(g^m(\boldsymbol{x}^m)))$. The expected predictive distribution for a sample $\boldsymbol{x} \in \mathcal{X}$ can be estimated from the predictive posterior $p(y \mid \boldsymbol{x}; \mathcal{D})$, which can be expressed from a Bayesian perspective as:

$$p(y \mid \boldsymbol{x}, \mathcal{D}) = \int_{f \in \mathcal{F}} p(y \mid \boldsymbol{x}, f)\, p(f \mid \mathcal{D})\, df, \tag{1}$$

where $\mathcal{F}$ denotes the space of predictors. For a given finite set of $M$ diverse models, this integral is typically approximated by a sum over the individual models (Lakshminarayanan et al., 2017):

$$p(y \mid \boldsymbol{x}, \mathcal{D}) \approx \sum_{m=1}^{M} p(y \mid \boldsymbol{x}^m, f^m)\, p(f^m \mid \mathcal{D}). \tag{2}$$

By Bayes' rule, $p(f^m \mid \mathcal{D}) = \frac{p(\mathcal{D}|f^m)\,p(f^m)}{p(\mathcal{D})}$. Under a uniform prior over the $M$ models (i.e., not favoring any modality in advance), i.e., $p(f^m) = \frac{1}{M}$, this reduces to $p(f^m \mid \mathcal{D}) = \frac{p(\mathcal{D}|f^m)}{\sum_{j=1}^{M} p(\mathcal{D}|f^j)}$. Substituting this into Eq. (2) yields a likelihood-weighted sum across modalities:

$$p(y \mid \boldsymbol{x}, \mathcal{D}) \approx \sum_{m=1}^{M} w_m\, p(y \mid \boldsymbol{x}^m, f^m), \tag{3}$$

where the weights satisfy $w_m \propto p(\mathcal{D} \mid f^m)$ and $\sum_{m=1}^{M} w_m = 1$. In particular, each $w_m$ can be expressed in terms of the cross-entropy loss (equivalently, the negative log-likelihood) that model $f^m$ exhibits on the dataset $\mathcal{D}$, i.e.,

$$w_m = \frac{\exp\big(-\mathrm{CE}(f^m, \mathcal{D})\big)}{\sum_{j=1}^{M} \exp\big(-\mathrm{CE}(f^j, \mathcal{D})\big)}, \tag{4}$$

where $\mathrm{CE}(f^m, \mathcal{D})$ denotes the average cross-entropy loss of model $f^m$ on $\mathcal{D}$.

Our goal is to capture the full predictive capability and uncertainty expressed in Eq. (3) to be reflected in the final classifier $h$, such that it inherits both the diverse uncertainty knowledge present at the logit level of each unimodal model and the strong discriminative power from the feature level. For convenience, we denote $p(y \mid \boldsymbol{x}, \mathcal{D})$ as $\boldsymbol{p}^{\mathrm{T}}$ and $p(y \mid \boldsymbol{x}^m, f^m)$ as $\boldsymbol{p}^m$. Accordingly, $h$ is trained by minimizing the cross-entropy with soft targets given by $\boldsymbol{p}^{\mathrm{T}}$. From the perspective of knowledge distillation (KD), the output $\boldsymbol{p}^{\mathrm{S}}$ of $h$ can be conceptually regarded as the student distribution while, $\boldsymbol{p}^{\mathrm{T}}$ serves as the teacher distribution. Consequently, this corresponds to minimizing the KL divergence between teacher and student distributions[1]:

$$\mathcal{L}_{\mathrm{KL}} = D_{\mathrm{KL}}(\boldsymbol{p}^{\mathrm{T}} \,\|\, \boldsymbol{p}^{\mathrm{S}}). \tag{5}$$

Notably, this loss formulation given by Eq. (5) and Eq. (3), accounts for the effect of underperforming unimodal models when training $h$ by ranking the teachers rather than treating them uniformly. To practically implement $w_m$, the weights are estimated on each mini-batch $\mathcal{B}$, as a stochastic approximation to $\mathcal{D}$.

We further decompose the KL divergence, following Zhao et al. (2022), into target and non-target class components, to gain deeper insights and better leverage the dark knowledge contained in the multimodal teacher logits for OOD detection. We therefore reformulate Eq. (5) as a weighted sum of these two components (the derivation is deferred to Section A.1):

$$\mathcal{L}_{\mathrm{KL}} = D_{\mathrm{KL}}\big(\boldsymbol{b}^{\mathrm{T}} \,\|\, \boldsymbol{b}^{\mathrm{S}}\big) + \big(1 - p_t^{\mathrm{T}}\big)\, D_{\mathrm{KL}}\big(\hat{\boldsymbol{p}}^{\mathrm{T}} \,\|\, \hat{\boldsymbol{p}}^{\mathrm{S}}\big), \tag{6}$$

where the notation $\boldsymbol{b} = [\,p_t,\, p_{\backslash t}\,] \in \mathbb{R}^2$ denotes the binary probabilities of the target class ($p_t$) and all non-target classes ($p_{\backslash t} = 1 - p_t$). The notation $\hat{\boldsymbol{p}} = [\,\hat{p}_1, \dots, \hat{p}_{t-1}, \hat{p}_{t+1}, \dots, \hat{p}_C\,] \in \mathbb{R}^{(C-1)}$ is used to independently model probabilities among non-target classes, i.e., the probabilities normalized excluding the $t$-th class.

Specifically, in Eq. (6), the first term transfers knowledge via binary logit distillation for the target class, referred to as target class KD (TCKD). The second term, referred to as non-target class KD (NCKD), considers the knowledge among the non-target logits. However, these two KD terms are coupled with $1 - p_t^{\mathrm{T}}$ in Eq. (6). To better adjust the importance of each KD term, the hyperparameters $\alpha$ and $\beta$ are introduced in Eq. (7), following Zhao et al. (2022). We further provide insights into each of these KD losses in the next section. Finally, we adopt the standard cross-entropy loss $\mathcal{L}_{\mathrm{CE}}$ as the task loss, and compute the final objective for training the classifier $h$ as:

$$\mathcal{L} = \mathcal{L}_{\mathrm{CE}} + \alpha \underbrace{D_{\mathrm{KL}}\big(\boldsymbol{b}^{\mathrm{T}} \,\|\, \boldsymbol{b}^{\mathrm{S}}\big)}_{\text{TCKD}} + \beta \underbrace{D_{\mathrm{KL}}\big(\hat{\boldsymbol{p}}^{\mathrm{T}} \,\|\, \hat{\boldsymbol{p}}^{\mathrm{S}}\big)}_{\text{NCKD}}. \tag{7}$$

In particular, the teachers' guidance itself is integrated into the joint classification head that operates on the combined embedding space of modalities, rather than relying on a separate student model

---

[1]We omit the temperature $\tau$ from Hinton et al. (2015) without loss of generality, to allow the student to better imitate the predictive posterior and to keep it in its natural form ($\tau = 1$), thereby preserving its diversity (Zhang et al., 2019; Wang et al., 2023a).

that would otherwise need to learn low-level perception of modalities. Overall, it leverages self-generated, uncertainty-aware targets—extracted from the logit space of multimodal models—to effectively learn holistic uncertainty knowledge across modalities. Therefore, the proposed framework harnesses the intrinsic OOD detection capability of a given set of multimodal models by exploiting both feature-level and logit-level information across the modalities.

### 3.4 Further Insights into Uncertainty-aware Dark Knowledge

To gain deeper insights into how the uncertainty-aware dark knowledge contained in multimodal teacher logits contributes to OOD detection, we reformulate Eq. (5) as the combination of TCKD and NCKD. In particular, Zhao et al. (2022) study their effect in the task of image classification. In contrast, below we investigate the role of these two forms of dark knowledge for multimodal OOD detection.

**Multimodal near-OOD detection mostly benefits from TCKD.** TCKD transfers knowledge about the relative "difficulty" of training samples, i.e., the knowledge describes how difficult it is to recognize each sample (Zhao et al., 2022). In our task, TCKD can be interpreted as binary uncertainty-aware supervision of the target class, which guides the student to identify difficult ID samples that lie near the decision boundary (i.e., with high uncertainty in the target class). This is particularly relevant to near-OOD detection, where the fundamental challenge is that near-OOD samples also lie close to ID decision boundaries (e.g, given 'cat' and 'dog' as ID classes, images of 'fox' will be near-OOD, see Fig. 3) and often exhibit ambiguity with those ID samples. Therefore, we argue that the near-OOD task mostly benefits from TCKD rather than NCKD (which focuses on the uncertainty knowledge among non-target classes) in the proposed self multimodal OOD distillation framework. This claim is further validated by the empirical results presented in Table 1, where we individually study the effects of TCKD and NCKD on two near-OOD benchmarks. The results imply that applying TCKD alone is comparable to, and even better than, classical KD (which combines both TCKD and NCKD, with results shown in the second row) for the near-OOD task.

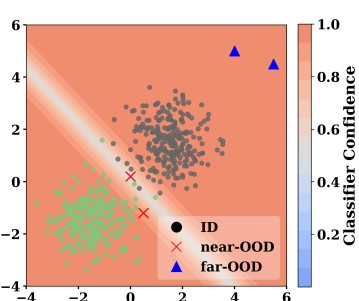

Figure 3: Classifier-based confidence for OOD detection. It assigns low scores in the small in-between area but may still suffer from overconfidence issues.

Table 1: The effect of TCKD and NCKD for near-OOD detection. Values in brackets represent the performance improvement over the baseline.

| TCKD | NCKD | HMDB51 25/26 | | | UCF101 50/51 | | |
|---|---|---|---|---|---|---|---|
| | | FPR95 | AUROC | ID ACC | FPR95 | AUROC | ID ACC |
| – | – | 38.78 | 88.83 | 89.76 | 10.10 | 98.06 | 99.71 |
| ✓ | ✓ | 34.81 (-3.97) | 90.07 (+1.24) | 90.37 (+0.61) | 5.98 (-4.12) | 98.57 (+0.51) | 99.79 (+0.08) |
| – | ✓ | 35.82 (-2.96) | 89.69 (+0.86) | 89.89 (+0.13) | 6.76 (-3.34) | 98.49 (+0.43) | 99.63 (-0.08) |
| ✓ | – | 34.99 (-3.79) | 90.12 (+1.29) | 90.50 (+0.74) | 5.92 (-4.18) | 98.57 (+0.51) | 99.81 (+0.10) |

**Multimodal Far-OOD detection mostly benefits from NCKD.** In contrast to near-OOD samples that reside close to decision boundaries, far-OOD samples lie far from the ID clusters or centers and therefore are less influenced by TCKD. However, a common challenge in the far-OOD case is that classifiers can still be overconfident in these sparse far-OOD regions (Fig. 3)—while they assign low confidence primarily near decision boundaries, they often misclassify far-OOD samples as belonging to one of the ID classes (Park et al., 2023). This overconfidence can be attributed to the fact that the model is trained with one-hot labels, which compel it to treat even atypical ID samples as strong representatives of their target classes, under the provided high-confidence supervision (Yang & Xu, 2025). Therefore, rather than relying on the binary uncertainty knowledge provided by TCKD, which is still related to the target class, the student benefits from NCKD on these ID samples, which emphasizes uncertainty knowledge among non-target classes. To further validate this claim, we evaluate the individual effects of TCKD and NCKD on four far-OOD datasets in Table 2. The results show that applying NCKD alone, or assigning more weight to NCKD, can

Table 2: The effect of TCKD and NCKD for far-OOD detection, on HMDB51 (Kuehne et al., 2011) ID dataset. Values in brackets represent the performance improvement over the baseline.

| TCKD | NCKD | UCF101 | | HAC | | EPIC-Kitchens | | Kinetics600 | |
|------|------|--------|------|------|------|------|------|------|------|
| | | FPR95 | AUROC | FPR95 | AUROC | FPR95 | AUROC | FPR95 | AUROC |
| – | – | 31.77 | 90.55 | 23.49 | 94.20 | 22.12 | 94.87 | 26.91 | 93.53 |
| ✓ | ✓ | 29.49 (-2.28) | 90.80 (+0.25) | 21.19 (-2.30) | 94.77 (+0.57) | 18.91 (-3.21) | 95.13 (+0.26) | 22.49 (-4.42) | 94.70 (+1.17) |
| ✓ | – | 29.83 (-1.94) | 90.70 (+0.15) | 21.48 (-2.01) | 94.71 (+0.51) | 19.59 (-2.53) | 94.96 (+0.09) | 22.81 (-4.10) | 94.64 (+1.11) |
| – | ✓ | 28.60 (-3.17) | 91.46 (+0.91) | 20.75 (-2.74) | 94.94 (+0.74) | 16.01 (-6.11) | 96.03 (+1.16) | 22.17 (-4.74) | 94.83 (+1.30) |

yield better detection performance (with gains of up to 15% on EPIC-Kitchens) on the multimodal far-OOD task.

**Our method improves OOD detection, which cannot be achieved by the joint probability distribution or by naively combining it with predictions from individual modalities.** Our method significantly improves both near-OOD and far-OOD detection by reducing overconfidence and false positive rates through learning with self uncertainty-aware dark knowledge distilled from the teachers. Notably, this performance gain (see Section 4.3 for empirical validation) cannot be achieved by relying on inference from the joint probability distribution across all modalities, $p^{\mathrm{S}}$ (trained solely with the task loss), nor by averaging the predicted probabilities of the individual modalities and the joint classifier: $\bar{p}_{\mathrm{all}} = \frac{1}{M+1}\left(\sum_{m=1}^{M} p^m + p^{\mathrm{S}}\right)$.

# 4 EVALUATION

In this section, we first describe our experimental setup, then present the main results on diverse multimodal OOD detection benchmarks, followed by ablation studies.

## 4.1 EXPERIMENTAL SETTINGS

**Datasets and Tasks.** Following (Dong et al., 2024; Li et al., 2025; Liu et al., 2025), we evaluate the proposed method on a diverse set of multimodal OOD benchmarks, comprising five action recognition datasets (HMDB51 (Kuehne et al., 2011), UCF101 (Soomro et al., 2012), Kinetics-600 (Kay et al., 2017), HAC (Dong et al., 2023), and EPIC-Kitchens (Damen et al., 2018)) and two tasks, namely multimodal near-OOD detection and multimodal far-OOD detection. HMDB51 and UCF101 provide video and optical flow modalities, whereas the others additionally include audio modality. For the near-OOD detection task, we evaluate on four datasets: EPIC-Kitchens 4/4, a subset of EPIC-Kitchens divided into four classes for training (ID) and four classes for testing (OOD); HMDB51 25/26, UCF101 50/51, and Kinetics-600 129/100, which are similarly derived from HMDB51, UCF101, and Kinetics-600, respectively. For the far-OOD detection task, either HMDB51 or Kinetics-600 is used as the ID dataset, with the remaining datasets serving as OOD.

**Evaluation Metrics.** We use widely adopted metrics for OOD detection (Dong et al., 2024; Li et al., 2025), including the area under the receiver operating characteristic curve (AUROC), the false positive rate at 95% true positive rate (FPR95), and the ID classification accuracy (ID ACC).

**Baselines.** As the vanilla baseline model (Base), we train all classifiers, including each unimodal model and a combined classifier (with the same architecture as ours), using only the task loss, i.e., cross-entropy loss. Following (Dong et al., 2024; Li et al., 2025; Liu et al., 2025), we adopt the SlowFast network (Feichtenhofer et al., 2019), initialized with pre-trained weights from Kinetics-600. As an easy plug-and-play approach, we further evaluate our method on recent SOTA models as backbones that incorporate retraining approaches, including the Agree-to-Disagree algorithm (A2D) (Dong et al., 2024), the combination of A2D and NP-Mix (AN) (Dong et al., 2024), DPU (Li et al., 2025), and Feature Mixing (FM) (Liu et al., 2025). For a fair comparison, we use the same classifier architecture as theirs, but retrain them using our framework while keeping the backbone models fixed.

**Configuration.** As mentioned, our method requires no retraining of the original unimodal models and involves only retraining a combined classifier. We use the same model architectures as in the baselines above and retrain their final classifiers using the proposed framework for 10 epochs. Training is performed with the Adam optimizer (Kingma, 2014), an initial learning rate of 0.0001,

Table 3: Multimodal near-OOD detection results using video and optical flow. The best results are bolded. Results averaged over six random runs.

| Method | HMDB51 25/26 | | | UCF101 50/51 | | | EPIC-Kitchens 4/4 | | | Kinetics600 129/100 | | |
|---|---|---|---|---|---|---|---|---|---|---|---|---|
| | FPR95 | AUROC | ID ACC | FPR95 | AUROC | ID ACC | FPR95 | AUROC | ID ACC | FPR95 | AUROC | ID ACC |
| Base | 38.78 | 88.83 | 89.76 | 10.10 | 98.06 | 99.71 | 75.00 | 66.70 | 71.27 | 64.61 | 76.59 | 80.31 |
| +Ours | **34.90** | **90.12** | **90.50** | **5.92** | **98.57** | **99.81** | **74.74** | **68.28** | **73.02** | **60.91** | **78.07** | **81.58** |
| A2D | 38.34 | 88.22 | 90.63 | 7.09 | 98.19 | 99.61 | **66.23** | **71.04** | 71.46 | 63.04 | 76.47 | 79.52 |
| +Ours | **37.65** | **88.81** | **90.81** | **5.40** | **98.49** | **99.71** | 68.62 | 70.33 | **72.35** | **61.42** | **77.77** | **81.35** |
| AN | **33.77** | 88.80 | 90.20 | 7.96 | 98.24 | 99.71 | 67.16 | 71.53 | 71.64 | 62.91 | 76.93 | 80.54 |
| +Ours | 34.07 | **90.03** | **90.46** | **5.51** | **98.56** | **99.77** | **65.67** | **72.50** | **72.01** | **61.16** | **78.06** | **82.29** |
| DPU | 34.42 | 89.15 | 92.16 | **7.57** | 98.17 | **99.81** | 63.81 | 71.46 | 72.39 | 61.59 | 77.50 | 81.07 |
| +Ours | **33.25** | **89.65** | **92.77** | 7.84 | **98.31** | 99.77 | 64.70 | 71.11 | **73.02** | **59.46** | **78.33** | **81.96** |
| FM | 45.10 | 87.29 | 89.11 | 8.06 | 97.92 | 99.61 | **71.83** | 68.49 | 72.76 | 64.10 | 76.16 | 80.11 |
| +Ours | **37.73** | **88.99** | **90.94** | **6.06** | **98.49** | **99.71** | 72.16 | **68.93** | **73.06** | **62.75** | **78.13** | **81.80** |

Table 4: Multimodal far-OOD detection results using video and optical flow, with HMDB51 as the ID dataset. The best results are bolded. Results averaged over six random runs.

| Method | Kinetics-600 | | UCF101 | | EPIC-Kitchens | | HAC | | Average | | ID ACC |
|---|---|---|---|---|---|---|---|---|---|---|---|
| | FPR95 | AUROC | FPR95 | AUROC | FPR95 | AUROC | FPR95 | AUROC | FPR95 | AUROC | |
| Base | 26.91 | 93.53 | 31.47 | 90.10 | 22.12 | 94.87 | 23.49 | 94.20 | 26.00 | 93.18 | 87.46 |
| +Ours | **22.17** | **94.83** | **28.60** | **91.46** | **16.01** | **96.03** | **20.75** | **94.94** | **21.88** | **94.31** | **88.12** |
| A2D | 20.18 | 95.12 | 33.87 | 90.29 | 12.43 | 96.53 | **15.85** | 95.82 | 20.58 | 94.44 | 87.34 |
| +Ours | **17.40** | **95.73** | **27.55** | **91.03** | **8.73** | **97.33** | 16.47 | **96.07** | **17.54** | **95.04** | **87.69** |
| AN | 24.29 | 93.99 | 36.94 | 89.71 | **7.18** | **97.60** | 23.15 | 94.45 | 22.89 | 93.94 | 86.66 |
| +Ours | **19.98** | **94.96** | **29.35** | **91.26** | 10.40 | 96.05 | **19.38** | **94.75** | **19.78** | **94.49** | **86.96** |
| DPU | 20.75 | **95.35** | 28.39 | **92.41** | **4.33** | **98.46** | 20.64 | **95.40** | 18.53 | **95.41** | 87.34 |
| +Ours | **19.11** | 95.32 | **25.25** | 92.26 | 6.50 | 97.66 | **18.15** | 95.36 | **17.25** | 95.16 | **87.94** |
| FM | 20.30 | **94.85** | 34.89 | 89.97 | **9.01** | **96.42** | 19.27 | **95.25** | 20.87 | **94.12** | **86.32** |
| +Ours | **16.88** | 94.79 | **27.99** | **90.64** | 12.03 | 93.32 | **16.88** | 94.94 | **18.95** | 93.92 | 86.26 |

cosine annealing learning-rate scheduler and a batch size of 16 on an NVIDIA-A100 GPU. Following (Zhao et al., 2022) we set the loss terms of KD and CE to 1.0 each. The hyperparameters are chosen as $\alpha = 0.1$, $\beta = 0.9$ for the far-OOD task, and $\alpha = 0.8$, $\beta = 0.2$ for the near-OOD task.

## 4.2 MAIN RESULTS

Results for the near-OOD detection task with all five datasets are presented in Table 3. Our method consistently achieves the best values across key metrics, including FPR95, AUROC, and ID accuracy on all baselines. Notably, simply incorporating our framework into a vanilla baseline (Base+Ours) yields substantial improvements in OOD performance, achieving comparable or even superior results to SOTA multimodal OOD detection methods that rely on complex retraining approaches.

**Our method is model-agnostic for diverse multimodal OOD detection methods.** As seen, across different training strategies, including A2D, AN, DPU, and FM, our method consistently improves performance on nearly every evaluation metric despite their diversity. For example, on the UCF101 dataset with AN, our method reduces FPR95 by up to 31%, while on HMDB51 with FM it achieves a reduction of 17%.

**Our method is effective in multiple OOD tasks.** Results for the far-OOD task with HMDB as the ID dataset are presented in Table 4, and with Kinetics-600 as the ID dataset in Table 6 in the appendix. Similar to the near-OOD task, our method yields considerable enhancements (with gains up to 30%) in OOD detection performance across all metrics and all baseline algorithms, demonstrating the effectiveness of leveraging uncertainty-aware dark knowledge for multimodal OOD detection.

**Performs robustly across diverse datasets.** Across five diverse datasets covering a wide range of video styles, including digitized movies, cartoon figures, everyday YouTube videos and kitchen environments, our method consistently reduces FPR95 and improves both AUROC and ID accuracy.

**Our method is adaptable and effective across various combinations of modalities,** not limited to video and optical flow. Results in Table 7 in the appendix and Table 4 show that performance is consistently improved regardless of the modality combinations used.

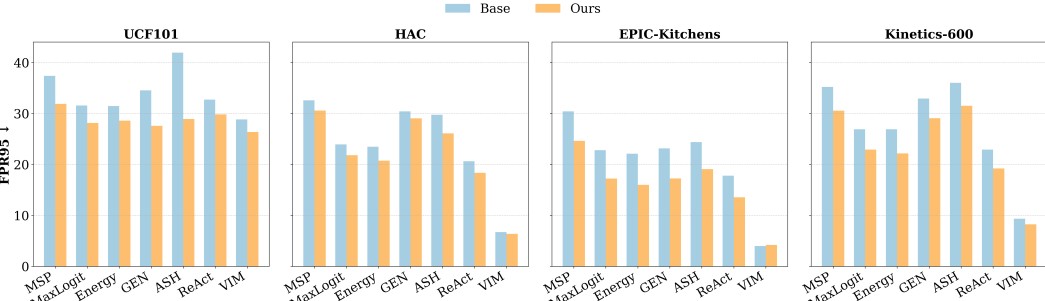

Figure 4: Compatibility of our method with various post-hoc OOD scoring methods.

## 4.3 ABLATION STUDIES

We conduct ablation studies with two main objectives: (1) to evaluate the compatibility of our method with various post-hoc OOD scores; and (2) to analyze the contribution of each individual component of our framework. A detailed hyperparameter analysis is provided in Section A.3.

**Compatibility with various post-hoc OOD scores.** We evaluate our method across seven post-hoc OOD scoring methods with different strategies: probability space (MSP), logit space (MaxLogit, Energy, GEN), penultimate activation manipulations (ReAct, ASH), and combinations of logit- and feature-space techniques (VIM), beyond the default model scores used in our main experiments. Results in Fig. 4 show that, despite this diversity, our method consistently improves the performance of all OOD scoring methods (with detailed results provided in Table 8 in the appendix).

**Importance of the components.** Our self multimodal OOD distillation framework improves OOD detection by leveraging uncertainty-aware dark knowledge from unimodal experts. To further validate this, we conduct the following experiments on four OOD datasets. First, we compare the performance of our method against a combined classifier $h$ trained using only the task loss (i.e., equivalent to setting both $\alpha$ and $\beta$ to 0 in Eq. (7), and denoted as *Vanilla* in Table 5). We also compare against an ensemble baseline that averages the predicted probabilities of the individual modalities and the joint classifier ($\bar{p}_{\text{all}}$, cf. Section 3.4), reported as *Ensemble* in Table 5. Our method surpasses these approaches, highlighting the importance of dark knowledge in improving OOD performance—something that cannot be achieved by naively combining ensembles or simply fusing features across modalities. We further evaluate a variant without modality-specific weights (in Eq. (3)), reported as *Uniform KD* in Table 5, which together demonstrates that our method effectively accounts for underperforming modalities. We also compare another knowledge transfer framework, where the $p^T$ is replaced with the best-performing modality while fusing features from all modalities, denoted as *Best KD* in Table 5. Our method still outperforms this variant, indicating that leveraging uncertainty-aware dark knowledge across all modalities is more effective than relying solely on the best individual modality.

Table 5: Ablation study of components for far-OOD detection with HMDB51 as the ID dataset. The best results are bolded. Results averaged over six random runs.

| Method | UCF101 | | HAC | | EPIC-Kitchens | | Kinetics-600 | | Average | | ID ACC |
|--------|--------|-------|-------|-------|---------------|-------|--------------|-------|---------|-------|--------|
| | FPR95 | AUROC | FPR95 | AUROC | FPR95 | AUROC | FPR95 | AUROC | FPR95 | AUROC | |
| Vanilla | 31.47 | 90.10 | 23.49 | 94.20 | 22.12 | 94.87 | 26.91 | 93.53 | 26.00 | 93.18 | 87.46 |
| Ensemble | 32.50 | 90.21 | 25.88 | 93.70 | 21.44 | 95.14 | 28.05 | 93.25 | 26.97 | 93.08 | 88.03 |
| Uniform KD | 29.44 | 91.26 | 22.37 | 94.62 | 16.62 | 95.87 | 23.40 | 94.58 | 22.96 | 94.08 | **88.21** |
| Best KD | 30.13 | 91.29 | 20.84 | 94.96 | 16.90 | 95.68 | 22.19 | 94.75 | 22.52 | 94.17 | 87.89 |
| Ours | **28.60** | **91.46** | **20.75** | **94.94** | **16.01** | **96.03** | **22.17** | **94.83** | **21.88** | **94.31** | 88.12 |

## 5 CONCLUSION

In this work, we explore the potential of dark knowledge within multimodal models to strengthen OOD detection. Building on this, we propose a self multimodal distillation framework that leverages both logit-space uncertainty knowledge and feature-space knowledge from a given set of multimodal models to harness their intrinsic OOD detection capability, while effectively accounting for underperforming modalities. Extensive experiments demonstrate that our method consistently improves multimodal OOD detection, further enhances existing approaches, and reveals the full synergy among modalities.

REPRODUCIBILITY STATEMENT

Our implementation strictly follows the benchmark guidelines provided in `https://github.com/donghao51/MultiOOD` (Dong et al., 2024). The settings and implementation details are reported in Section 4.1. Our code is publicly available at `https://github.com/codebyhdnu-hub/SMOD`. Detailed information on the hardware and software used is also provided in the repository.

THE USE OF LARGE LANGUAGE MODELS (LLMS)

We used ChatGPT (OpenAI, GPT-4 Turbo) solely for minor grammar and language corrections. All scientific content and analysis were entirely developed by the authors.

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

# A APPENDIX

## A.1 ADDITIONAL DETAILS OF THE METHOD

Following Zhao et al. (2022), we reformulate KL divergence as a weighted sum of two terms, in Eq. (6). A detailed derivation of Eq. (6) is provided below. For a given conditional distribution over classes, let the probability of the $i$-th class be denoted as $p_i$, i.e.,

$$p_i = \frac{\exp(z_i)}{\sum_{j=1}^{C} \exp(z_j)},$$

where $z_i$ is the logit corresponding to the $i$-th class. Then, we have $p_t$, $p_{\backslash t}$, and each element $\hat{p}_i$ of $\hat{\boldsymbol{p}}$ as (with notation consistent with Section 3.3):

$$p_t = \frac{\exp(z_t)}{\sum_{j=1}^{C} \exp(z_j)}, \quad p_{\backslash t} = \frac{\sum_{k=1,\,k\neq t}^{C} \exp(z_k)}{\sum_{j=1}^{C} \exp(z_j)}, \quad \hat{p}_i = \frac{\exp(z_i)}{\sum_{j=1,\,j\neq t}^{C} \exp(z_j)}. \tag{8}$$

$$\begin{aligned}
\mathcal{L}_{\mathrm{KL}} &= D_{\mathrm{KL}}(\boldsymbol{p}^{\mathrm{T}} \,\|\, \boldsymbol{p}^{\mathrm{S}}) \\
&= \sum_{i=1}^{C} p_i^{\mathrm{T}} \log\left(\frac{p_i^{\mathrm{T}}}{p_i^{\mathrm{S}}}\right) \\
&= p_t^{\mathrm{T}} \log\left(\frac{p_t^{\mathrm{T}}}{p_t^{\mathrm{S}}}\right) + \sum_{i=1,\,i\neq t}^{C} p_i^{\mathrm{T}} \log\left(\frac{p_i^{\mathrm{T}}}{p_i^{\mathrm{S}}}\right).
\end{aligned} \tag{9}$$

According to Eq. (8), we have $\hat{p}_i = p_i/p_{\backslash t}$; therefore, Eq. (9) can be rewritten as:

$$\begin{aligned}
\mathcal{L}_{\mathrm{KL}} &= p_t^{\mathrm{T}} \log\left(\frac{p_t^{\mathrm{T}}}{p_t^{\mathrm{S}}}\right) + \sum_{i=1,\,i\neq t}^{C} p_{\backslash t}^{\mathrm{T}} \hat{p}_i^{\mathrm{T}} \log\left(\frac{p_{\backslash t}^{\mathrm{T}} \hat{p}_i^{\mathrm{T}}}{p_{\backslash t}^{\mathrm{S}} \hat{p}_i^{\mathrm{S}}}\right) \\
&= p_t^{\mathrm{T}} \log\left(\frac{p_t^{\mathrm{T}}}{p_t^{\mathrm{S}}}\right) + \sum_{i=1,\,i\neq t}^{C} p_{\backslash t}^{\mathrm{T}} \hat{p}_i^{\mathrm{T}} \left(\log\left(\frac{\hat{p}_i^{\mathrm{T}}}{\hat{p}_i^{\mathrm{S}}}\right) + \log\left(\frac{p_{\backslash t}^{\mathrm{T}}}{p_{\backslash t}^{\mathrm{S}}}\right)\right) \\
&= p_t^{\mathrm{T}} \log\left(\frac{p_t^{\mathrm{T}}}{p_t^{\mathrm{S}}}\right) + \sum_{i=1,\,i\neq t}^{C} p_{\backslash t}^{\mathrm{T}} \hat{p}_i^{\mathrm{T}} \log\left(\frac{\hat{p}_i^{\mathrm{T}}}{\hat{p}_i^{\mathrm{S}}}\right) + \sum_{i=1,\,i\neq t}^{C} p_{\backslash t}^{\mathrm{T}} \hat{p}_i^{\mathrm{T}} \log\left(\frac{p_{\backslash t}^{\mathrm{T}}}{p_{\backslash t}^{\mathrm{S}}}\right),
\end{aligned} \tag{10}$$

Since $p_t^{\mathrm{T}}$ and $p_{\backslash t}^{\mathrm{S}}$ are independent of the class index $i$, we have:

$$\sum_{i=1,\,i\neq t}^{C} p_{\backslash t}^{\mathrm{T}} \hat{p}_i^{\mathrm{T}} \log\left(\frac{p_{\backslash t}^{\mathrm{T}}}{p_{\backslash t}^{\mathrm{S}}}\right) = p_{\backslash t}^{\mathrm{T}} \log\left(\frac{p_{\backslash t}^{\mathrm{T}}}{p_{\backslash t}^{\mathrm{S}}}\right) \sum_{i=1,\,i\neq t}^{C} \hat{p}_i^{\mathrm{T}} = p_{\backslash t}^{\mathrm{T}} \log\left(\frac{p_{\backslash t}^{\mathrm{T}}}{p_{\backslash t}^{\mathrm{S}}}\right). \tag{11}$$

Then, from Eq. (10) and Eq. (11), we obtain

$$\mathcal{L}_{\mathrm{KL}} = \underbrace{p_t^{\mathrm{T}} \log\left(\frac{p_t^{\mathrm{T}}}{p_t^{\mathrm{S}}}\right) + p_{\backslash t}^{\mathrm{T}} \log\left(\frac{p_{\backslash t}^{\mathrm{T}}}{p_{\backslash t}^{\mathrm{S}}}\right)}_{D_{\mathrm{KL}}(\boldsymbol{b}^{\mathrm{T}} \,\|\, \boldsymbol{b}^{\mathrm{S}})} + p_{\backslash t}^{\mathrm{T}} \underbrace{\sum_{i=1,\,i\neq t}^{C} \hat{p}_i^{\mathrm{T}} \log\left(\frac{\hat{p}_i^{\mathrm{T}}}{\hat{p}_i^{\mathrm{S}}}\right)}_{D_{\mathrm{KL}}(\hat{\boldsymbol{p}}^{\mathrm{T}} \,\|\, \hat{\boldsymbol{p}}^{\mathrm{S}})}. \tag{12}$$

It can be seen that Eq. (12) is a combination of two KL divergence loss terms, which can be rewritten as follows, identical to Eq. (6) in Section 3.3:

$$\mathcal{L}_{\mathrm{KL}} = D_{\mathrm{KL}}\left(\boldsymbol{b}^{\mathrm{T}} \,\|\, \boldsymbol{b}^{\mathrm{S}}\right) + \left(1 - p_t^{\mathrm{T}}\right) D_{\mathrm{KL}}\left(\hat{\boldsymbol{p}}^{\mathrm{T}} \,\|\, \hat{\boldsymbol{p}}^{\mathrm{S}}\right).$$

## A.2 ADDITIONAL RESULTS

We provide additional experimental results in this section. Table 6 presents the performance of our method on far-OOD detection with Kinetics-600 as the ID dataset. The results show consistent

Table 6: Multimodal far-OOD detection results using video and optical flow, with Kinetics-600 as the ID dataset. The best results are bolded. Results averaged over six random runs.

| Method | UCF101 | | HAC | | EPIC-Kitchens | | HMDB51 | | Average | | ID ACC |
|---|---|---|---|---|---|---|---|---|---|---|---|
| | FPR95 | AUROC | FPR95 | AUROC | FPR95 | AUROC | FPR95 | AUROC | FPR95 | AUROC | |
| Base | 70.67 | 68.49 | 55.43 | 78.40 | 37.93 | 85.10 | 66.08 | 68.80 | 57.53 | 75.20 | 73.71 |
| +Ours | **59.60** | **76.37** | **44.86** | **83.68** | **26.58** | **90.10** | **59.96** | **77.93** | **47.75** | **82.02** | **75.24** |
| A2D | 71.47 | 67.99 | 56.53 | 78.18 | 39.87 | 83.96 | 67.30 | 67.98 | 58.79 | 74.03 | 73.61 |
| +Ours | **59.55** | **75.97** | **43.06** | **84.55** | **26.89** | **90.29** | **60.70** | **77.51** | **47.55** | **82.08** | **75.83** |
| AN | 67.17 | 74.49 | 56.69 | 80.20 | 34.12 | 87.49 | 63.24 | 74.13 | 55.31 | 79.08 | 73.65 |
| +Ours | **58.40** | **79.08** | **42.67** | **85.34** | **28.27** | **90.15** | **60.56** | **80.22** | **47.48** | **83.70** | **76.14** |
| DPU | **55.33** | **78.20** | 47.39 | 82.99 | 27.38 | **91.61** | 61.27 | **80.83** | 47.84 | **83.91** | 76.74 |
| +Ours | 57.69 | 75.87 | **45.30** | **83.42** | **26.66** | 91.19 | **60.41** | 79.82 | **47.52** | 82.58 | **77.74** |

Table 7: Multimodal Near-OOD Detection using video, optical flow, and audio on Kinetics-600 (129/100). Each cell reports baseline / Ours, with the better value in bold. Results are compared across various post-hoc OOD scoring methods.

| Method | Base | | | AN | | | DPU | | |
|---|---|---|---|---|---|---|---|---|---|
| | FPR95 | AUROC | ID ACC | FPR95 | AUROC | ID ACC | FPR95 | AUROC | ID ACC |
| MSP | 61.73 / **60.40** | 77.24 / **78.97** | 80.46 / **82.15** | 58.85 / **57.96** | 78.50 / **79.41** | 81.76 / **83.13** | 63.34 / **61.57** | 77.47 / **78.26** | 82.68 / **83.33** |
| MaxLogit | 62.81 / **61.40** | 78.07 / **79.39** | 80.46 / **82.15** | 60.98 / **59.59** | 78.61 / **79.54** | 81.76 / **83.13** | 65.87 / **64.77** | 77.86 / **78.50** | 82.68 / **83.33** |
| Energy | 63.06 / **61.61** | 77.72 / **78.90** | 80.46 / **82.15** | 60.91 / **59.65** | 78.17 / **79.00** | 81.76 / **83.13** | 66.05 / **64.97** | 77.57 / **78.13** | 82.68 / **83.33** |
| GEN | 61.75 / **60.59** | 77.80 / **78.73** | 80.46 / **82.15** | 59.22 / **58.94** | 78.61 / **79.16** | 81.76 / **83.13** | 65.85 / **65.20** | 77.17 / **77.82** | 82.68 / **83.33** |
| ASH | 62.34 / **59.83** | 78.46 / **79.31** | 80.13 / **81.66** | 58.94 / **58.18** | 79.05 / **79.80** | 81.11 / **82.68** | 64.85 / **63.59** | 77.31 / **78.29** | 81.86 / **82.74** |
| ReAct | 69.05 / **65.73** | 75.76 / **77.39** | 80.44 / **82.11** | 66.85 / **64.18** | 76.40 / **77.44** | 81.86 / **82.72** | 69.01 / **67.93** | 76.77 / **77.13** | 82.39 / **83.05** |
| VIM | 62.97 / **61.81** | 77.73 / **78.90** | 80.46 / **82.15** | 61.06 / **59.69** | 78.19 / **78.79** | 81.76 / **83.13** | 65.81 / **65.42** | 77.79 / **78.22** | 82.68 / **83.33** |

improvements over all baselines across four diverse OOD datasets. These results highlight the effectiveness of our method in detecting far-OOD samples. Table 7 shows the near-OOD detection results on Kinetics-600 using three input modalities: video, optical flow, and audio. It also compares the performance of various post-hoc OOD scoring methods for each baseline in this setting. The results demonstrate that our method remains adaptable and effective across diverse modality combinations.

As discussed, we evaluate our method across seven post-hoc OOD scoring methods with different strategies: probability space (MSP), logit space (MaxLogit, Energy, GEN), penultimate activation manipulations (ReAct, ASH), and combined logit–feature space techniques (VIM). Detailed results on all baselines across four diverse datasets are provided in Table 8, with a summary in Fig. 5. The results demonstrate that, despite this diversity, our method consistently improves the performance of all OOD scoring methods.

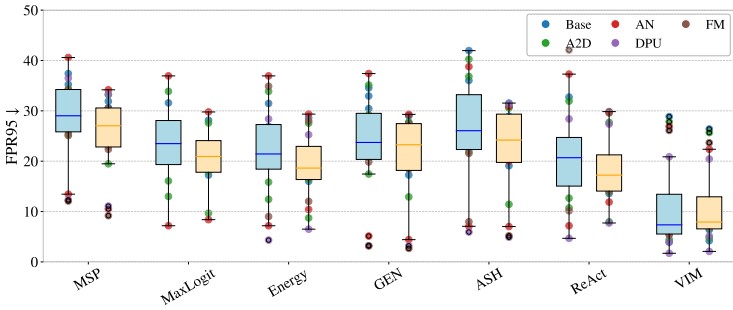

Figure 5: Summary of the performance of our method for far-OOD detection using video and optical flow, with HMDB51 as the ID dataset, evaluated across different models and various post-hoc OOD scoring methods.

Table 8: Comparison of different post-hoc OOD scoring methods for multimodal far-OOD detection using video and optical flow, with HMDB51 as the ID dataset. Each cell reports baseline / Ours, with the better value in bold.

| | Method | UCF101 FPR95 ↓ | UCF101 AUROC ↑ | HAC FPR95 ↓ | HAC AUROC ↑ | EPIC-Kitchens FPR95 ↓ | EPIC-Kitchens AUROC ↑ | Kinetics-600 FPR95 ↓ | Kinetics-600 AUROC ↑ |
|---|---|---|---|---|---|---|---|---|---|
| Base | MSP | 37.40 / **31.90** | 88.68 / **91.33** | 32.61 / **30.60** | 89.76 / **91.49** | 30.44 / **24.63** | 90.88 / **93.67** | 35.23 / **30.56** | 89.16 / **91.41** |
| | MaxLogit | 31.58 / **28.14** | 90.22 / **92.21** | 23.95 / **21.82** | 93.98 / **94.71** | 22.81 / **17.24** | 94.63 / **95.92** | 26.91 / **22.92** | 93.29 / **94.58** |
| | Energy | 31.47 / **28.60** | 90.10 / **91.46** | 23.49 / **20.75** | 94.20 / **94.94** | 22.12 / **16.01** | 94.87 / **96.03** | 26.91 / **22.17** | 93.53 / **94.83** |
| | GEN | 34.55 / **27.59** | 90.94 / **92.65** | 30.44 / **29.05** | 91.93 / **92.70** | 23.15 / **17.26** | 94.13 / **95.58** | 32.95 / **29.08** | 91.13 / **92.53** |
| | ASH | 41.96 / **28.94** | 88.97 / **91.77** | 29.76 / **26.11** | 93.21 / **94.25** | 24.40 / **19.09** | 94.16 / **95.63** | 36.03 / **31.54** | 91.23 / **92.93** |
| | ReAct | 32.73 / **29.85** | 90.19 / **91.08** | 20.64 / **18.36** | 95.26 / **95.34** | 17.79 / **13.57** | 95.65 / **96.29** | 22.92 / **19.22** | 94.73 / **95.40** |
| | VIM | 28.85 / **26.39** | 91.97 / **92.30** | 6.73 / **6.39** | **98.75** / 98.69 | **3.99** / 4.20 | 99.01 / **99.12** | 9.35 / **8.26** | 98.12 / **98.29** |
| A2D | MSP | 34.32 / **33.23** | 89.07 / **90.82** | **26.11** / 28.83 | 91.27 / **91.98** | 25.31 / **19.48** | 92.94 / **95.14** | 31.36 / **29.60** | 90.61 / **91.84** |
| | MaxLogit | 33.87 / **27.59** | 90.46 / **91.69** | **16.08** / 17.99 | 95.61 / **95.75** | 13.00 / **9.67** | 96.41 / **97.36** | 20.41 / **18.56** | 94.93 / **95.47** |
| | Energy | 33.87 / **27.55** | 90.29 / **91.03** | **15.85** / 16.47 | 95.82 / **96.07** | 12.43 / **8.73** | 96.53 / **97.33** | 20.18 / **17.40** | 95.12 / **95.73** |
| | GEN | 35.12 / **27.80** | 91.89 / **93.10** | **22.23** / 23.99 | 93.69 / **94.10** | 17.45 / **12.91** | 95.83 / **96.95** | 29.19 / **25.93** | 92.74 / **93.51** |
| | ASH | 40.25 / **30.56** | 89.17 / **91.75** | 25.77 / **23.81** | 93.61 / **94.62** | 21.89 / **11.43** | 94.94 / **97.33** | 36.83 / **30.79** | 91.48 / **93.15** |
| | ReAct | 31.93 / **27.71** | 90.10 / **90.46** | **12.66** / 14.00 | **96.46** / 96.28 | 10.72 / **7.96** | 96.83 / **97.24** | 15.85 / **14.94** | 95.94 / **96.18** |
| | VIM | 27.82 / **25.68** | **90.72** / 90.67 | **5.93** / 6.61 | **98.72** / 98.68 | **4.45** / 4.79 | 98.68 / **98.72** | 7.98 / **7.84** | 98.10 / **98.20** |
| AN | MSP | 40.59 / **34.18** | 88.00 / **90.44** | 28.62 / **26.13** | 91.57 / **92.99** | 13.45 / **10.49** | 96.42 / **97.66** | 29.42 / **27.39** | 90.65 / **92.58** |
| | MaxLogit | 36.94 / **29.78** | 89.73 / **91.72** | 23.03 / **20.82** | 94.25 / **94.91** | **7.18** / 8.39 | **97.72** / 97.53 | 24.63 / **21.05** | 93.72 / **94.90** |
| | Energy | 36.94 / **29.35** | 89.71 / **91.26** | 23.15 / **19.38** | 94.45 / **94.75** | **7.18** / 10.40 | **97.60** / 96.05 | 24.29 / **19.98** | 93.99 / **94.96** |
| | GEN | 37.40 / **29.30** | 91.24 / **93.14** | 24.29 / **22.55** | 94.33 / **95.13** | 5.13 / **4.42** | 99.02 / **99.12** | 25.20 / **23.97** | 93.54 / **94.72** |
| | ASH | 38.77 / **30.90** | 89.54 / **92.06** | 23.72 / **19.98** | 94.47 / **95.42** | 7.07 / **7.02** | **98.24** / 98.21 | 27.59 / **24.58** | 93.19 / **94.53** |
| | ReAct | 37.29 / **29.71** | 89.69 / **90.92** | 20.75 / **17.17** | **95.05** / 94.78 | **7.18** / 11.88 | **97.46** / 95.35 | 21.44 / **17.42** | 94.76 / **95.11** |
| | VIM | 26.91 / **22.37** | 92.28 / **92.70** | **6.39** / 6.98 | **98.56** / 98.41 | **5.59** / 7.89 | **98.06** / 97.07 | 9.35 / **7.94** | 98.04 / **98.05** |
| DPU | MSP | 36.49 / **33.27** | 90.52 / **91.18** | 27.37 / **25.95** | 93.30 / **93.61** | 12.31 / **11.08** | 97.52 / **97.67** | 27.59 / **26.68** | 93.17 / **93.37** |
| | Energy | 28.39 / **25.25** | **92.41** / 92.26 | 20.64 / **18.15** | **95.40** / 95.36 | **4.33** / 6.50 | **98.46** / 97.66 | 20.75 / **19.11** | **95.35** / 95.32 |
| | GEN | 28.16 / **25.47** | 93.53 / **94.17** | 21.44 / **19.04** | 95.66 / **96.05** | **3.19** / **3.19** | **99.30** / 99.27 | 21.09 / **20.66** | 95.45 / **95.66** |
| | ASH | 32.27 / **31.36** | 91.86 / **92.57** | 22.46 / **21.05** | 95.23 / **95.59** | 5.93 / **4.90** | **99.02** / 99.01 | 26.34 / **25.27** | 94.53 / **94.79** |
| | ReAct | 28.39 / **27.37** | **91.95** / 91.50 | 18.02 / **17.29** | **95.78** / 95.28 | **4.68** / 7.73 | **98.28** / 97.22 | 18.02 / **17.77** | **95.78** / 95.30 |
| | VIM | **20.87** / 20.43 | **94.42** / 94.17 | **3.88** / 5.13 | **99.16** / 98.98 | **1.71** / 2.08 | **99.63** / 99.48 | **6.27** / 6.84 | **98.74** / 98.57 |
| FM | MSP | 34.21 / **30.44** | 89.35 / **91.28** | 25.09 / **22.35** | 92.39 / **93.42** | 12.09 / **9.18** | 97.14 / **97.89** | 26.00 / **22.98** | 92.33 / **93.79** |
| | Energy | 34.89 / **27.99** | 89.97 / **90.64** | 19.27 / **16.88** | **95.25** / 94.94 | **9.01** / 12.03 | **96.42** / 93.32 | 20.30 / **16.88** | **94.85** / 94.79 |
| | GEN | 28.85 / **27.42** | 92.11 / **93.18** | 19.84 / **18.59** | 95.09 / **95.39** | 3.19 / **2.68** | 99.21 / **99.42** | 20.52 / **18.47** | 95.11 / **95.82** |
| | ASH | 32.27 / **28.45** | 90.08 / **92.31** | 21.55 / **20.75** | 94.25 / **94.99** | 7.98 / **5.19** | 98.21 / **98.57** | 23.38 / **20.35** | 93.74 / **95.07** |
| | ReAct | 42.08 / **29.59** | 88.39 / **89.89** | 22.23 / **16.53** | **94.79** / 94.77 | **10.15** / 14.08 | **95.42** / 92.24 | 23.49 / **16.48** | 94.32 / **94.56** |
| | VIM | 26.11 / **23.66** | **92.36** / 92.19 | 8.78 / **8.67** | **98.24** / 98.03 | **5.36** / 9.92 | **97.96** / 96.43 | 10.95 / **10.43** | **97.54** / 97.35 |

## A.3 HYPERPARAMETER ANALYSIS

In Fig. 6, we investigate the impact of the hyperparameters $\alpha$ and $\beta$ in our method across three far-OOD datasets. Following the basic settings in Zhao et al. (2022); Park et al. (2019), we set the loss weights of the KD and CE terms to 1.0. As discussed in Section 3.4, assigning a higher value to $\beta$, which emphasizes the NCKD component, leads to improved detection performance in multimodal far-OOD detection. Furthermore, the detection performance consistently remains above the baseline across all tested combinations of $\alpha$ and $\beta$, demonstrating the stability and robustness of our framework under varying hyperparameter settings.

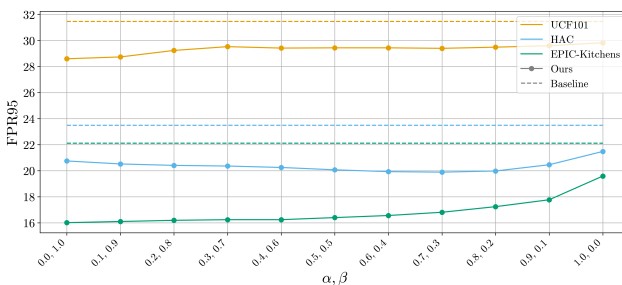

Figure 6: OOD detection performance with varying $\alpha$ and $\beta$ for far-OOD detection using video and optical flow, with HMDB51 as the ID dataset.

For real-world deployment where the type of OOD shift is unknown, we recommend a balanced and task-agnostic configuration such as $\alpha = 0.5$ and $\beta = 0.5$, which performs robustly across all benchmarks and OOD scenarios. Alternatively, practitioners may adopt a proxy-OOD tuning strategy, which is conventional in prior OOD detection (Hendrycks et al., 2019; Dong et al., 2024; Zhang et al., 2023). In practice, many real-world applications naturally prioritize one type of OOD. Some systems primarily focus on near-OOD detection (e.g., fine-grained classification, medical imaging, product or defect inspection) (Zhang et al., 2023), where far-OOD samples are trivial and typically filtered earlier in the pipeline. Conversely, some applications prioritize far-OOD (e.g., OCR scanners detecting non-text inputs need to be rejected early in pipeline). Our framework allows practitioners to adjust $\alpha$ and $\beta$ according to such priorities, but importantly, no prior knowledge of the exact OOD type is required to benefit from the method, as it consistently improves both near- and far-OOD detection under a wide range of settings.

