# OpenReview forum: "Leveraging Dark Knowledge for Intrinsic Multimodal Out-of-Distribution Detection"
_ICLR.cc/2026/Conference — Submitted to ICLR 2026_

### Official Review · Reviewer_XVtv · 2025-10-27

**Soundness:** 2
**Presentation:** 1
**Contribution:** 2
**Rating:** 4
**Confidence:** 5

**Summary:**

This work focuses on multi-modality OOD detection with semantic shift. Particularly, it employs a teach-student knowledge distillation framework to perform OOD detection. Specifically, the teacher is constructed as ensemble of pre-trained unimodal classifiers, and the soft targets are calculated as the weighted predictive targets from  pre-trained unimodal classifiers. The corresponding weight for each modality is calculated as modality-specific CE loss over the sum of CE loss across all modalities. The final loss incorporates a standard cross entropy loss and the classical decoupled knowledge distillation loss.

**Strengths:**

Strengths:
- The paper is well-written at a high level, and the proposed framework sounds reasonable.
- It is appreciated that the authors conduct experiments regarding the contribution of target class knowledge distillation (TCKD) and non-target-class knowledge distillation (TCKD) to multi-modal OOD detection.
- The experiments are comprehensive.

**Weaknesses:**

Weaknesses
-  The paper is easy to follow at the first glance, but several details are missing when read more.
   - Figure 2 can be further improved by adding the symbols for each block to improve clarity;
   - It seems to the feature fusion is implemented by concatenating them directly, correct me if i am wrong. Otherwise, please clarify it in Line 166;
   - The performance gain is somehow marginal. Particularly, it can be seen from Table 4 that DPU solely achieves the best performance in terms of AUROC.  Moreover, there is a trade-off between AUROC and FPR95 for DPU and FM.

-  The proposed framework is ok to do but I did not see a strong connection with mutli-modal OOD detection. To me, it's more like improving the accuracy of the joint classier, which can be empirically verified by the ID accuracy from Table 4.

**Questions:**

See weaknesses.

---

> ### Author Response · Authors · 2025-11-27
> **Response to Reviewer XVtv**
>
> We sincerely thank the reviewer for the constructive feedback and for acknowledging the overall clarity of our work, the reasonableness of the proposed framework, and the comprehensive experimental evaluation. We appreciate that the reviewer found value in our exploration of the contributions of target class knowledge distillation (TCKD) and non-target class knowledge distillation (NCKD) to multimodal OOD detection.
> Below, we address the concerns raised in the review:
>
> **Q1.** Figure 2 can be further improved by adding the symbols.\
> **A1.** Thank you for your helpful suggestion. We have added distinct symbols for each object in Figure 2 and updated the legend to enhance clarity and interpretability in the revised manuscript.
>
> **Q2.** Feature fusion is implemented by concatenating them directly?\
> **A2.** Yes, your understanding is correct---the modality features are fused through feature concatenation. We have revised Line 166 to explicitly clarify this by adding:
> $ \mathbf{z} = [\, \mathbf{z}^1 \, || \, \dots \, || \, \mathbf{z}^M \,],$ where $||$ denotes concatenation.
> We follow the earlier convention used in the MultiOOD benchmark (Dong et al., 2024) for formulating the multimodal problem. We appreciate your suggestion and have clarified this in the updated manuscript.
>
> **Q3.** Performance gain for DPU and FM.\
> **A3.** While it is correct that the DPU and FM baselines occasionally achieve slightly higher AUROC values for far-OOD detection in Table 4, we would like to emphasize that both methods demonstrate strong performance gains with our approach across all datasets except EPIC-Kitchens. Specifically, for the remaining three OOD benchmarks, our method consistently and significantly reduces FPR95 (by up to 12\% for DPU and up to 20\% for FM), while maintaining comparable AUROC performance. In cases where AUROC shows a slight decrease, the drop is marginal and not statistically significant, whereas the reductions in FPR95 are substantial and practically meaningful. Furthermore, ID ACC is maintained or slightly improved, confirming that our enhancement does not compromise ID performance. Notably, in the more challenging near-OOD setting (Table 3), our method achieves even stronger gains for these baselines, further highlighting its robustness.
>
> **Q4.** I did not see a strong connection with mutli-modal OOD detection.\
> **A4.** We thank the reviewer for raising this concern. While it is true that our framework also improves the accuracy of the joint classifier, our method is explicitly designed to improve OOD detection by leveraging the complementary nature of multiple modalities, rather than merely boosting ID accuracy.
> The standard approach for OOD detection is to derive a score function from a trained network such that ID samples exhibit higher scores than OOD samples. A major paradigm in designing this score function is to use the classifier’s output signals, often referred to as “confidence”. Classifier-based confidence scores are particularly advantageous for OOD detection, as they fully utilize class-dependent information of ID data and provide fine-grained detection capability (Park et al., 2023).
> However, a common issue is overconfidence in neural network classifiers: they may assign high confidence to far-OOD samples, incorrectly classifying them as ID, and may also misclassify hard or ambiguous ID samples near decision boundaries as OOD (Hendrycks \& Gimpel, 2017, Park et al., 2023).\
> Our method specifically aims to reduce this overconfidence by incorporating uncertainty-aware soft targets derived from multimodal experts, instead of relying on one-hot labels, which treat all ID samples uniformly as perfect class representatives.
> Through our self multimodal OOD distillation framework, we explicitly account for underperforming modalities and effectively transfer modality-aware uncertainty to the joint classifier, thereby reducing overconfidence and improving OOD detection.
> This approach can also maintain, or even improve, ID accuracy by reasonably learning ambiguous ID samples. It is also worth noting that the performance gains in OOD detection are substantially higher than those in ID accuracy, as shown in Section 4.2 of the manuscript.
>
> Thank you once again for your valuable feedback. We have carefully addressed all the comments and made revisions based on your suggestions, which we believe have greatly enhanced the quality and clarity of our paper.

---

### Official Review · Reviewer_qzLe · 2025-10-30

**Soundness:** 2
**Presentation:** 2
**Contribution:** 2
**Rating:** 2
**Confidence:** 4

**Summary:**

The core contribution of this paper is the application of two existing techniques (Decoupled Knowledge Distillation and Weighted Ensemble Distillation) to the specific task of multimodal OOD detection.
1. Decoupled KL Divergence: The $\mathcal{L}_{KL}$ loss function used for OOD detection, particularly its decomposition into Target Class Knowledge Distillation (TCKD) and Non-Target Class Knowledge Distillation (NCKD) is directly adopted from the work of (Zhao et al., 2022).
2. Dark Knowledge Analysis: The insights presented in Section 3.4, which claim that TCKD benefits near-OOD and NCKD benefits far-OOD detection , are also primarily derived from the findings of (Zhao et al., 2022).
3. Teacher Model: The paper's "teacher" distribution $p^T$ is obtained by a weighted average of the unimodal expert predictions (Eq. 3). This is, in essence, a standard ensemble distillation technique.

**Strengths:**

This paper is well-written and very easy to follow.

**Weaknesses:**

1. The paper claims its method is a "plug-and-play" framework that improves the performance of existing SOTA methods (like A2D, AN, DPU, FM). However, this comparison is problematic. The paper explicitly states that when combining with SOTA methods, it keeps the SOTA model backbones fixed and only retrains their final classifier for 10 epochs. The SOTA baselines (A2D, DPU, etc.) are complex strategies that involve end-to-end training of the backbone to learn "a more discriminative embedding space". Therefore, the results (e.g., "A2D" vs. "A2D + Ours" in Table 3 and 4) are actually comparing a "fully trained SOTA model" against a "fully trained SOTA model + additional classifier fine-tuning". At best, this comparison only proves that fine-tuning the classifier head with the proposed distillation loss can provide a marginal boost to an already-trained model.
2. The paper assumes that "poor ID classification performance" equates to "poor OOD detection capability," but provides no evidence to support this critical assumption. It is entirely possible for a modality (e.g., optical flow in Fig. 1 ) to have acceptable ID classification but poor OOD detection due to overconfidence. This weighting scheme may be penalizing modalities based on the wrong signal.
3. The paper explicitly states that the hyperparameters $\alpha$ and $\beta$ (which balance TCKD and NCKD) must be set differently depending on the task: for the far-OOD task, $\alpha=0.1, \beta=0.9$, while for the near-OOD task, $\alpha=0.8, \beta=0.2$. In a real-world application, the system cannot know in advance whether the OOD samples it will encounter are "near" or "far."
4.  The paper's title and text use the word "Intrinsic", claiming to mine the "intrinsic OOD detection capability." However, the method explicitly introduces and trains a new joint classifier $h$ and a new distillation loss $\mathcal{L}_{KL}$. This is not mining "inherent" capability; it is introducing new knowledge via fine-tuning.
5. The method is neither a truly training-free post-hoc scoring function like MSP or Energy , nor is it a full end-to-end training method. It is a classifier-head fine-tuning method that requires 10 epochs of training. So, I don't agree that this is a Post-hoc method.
6. The paper heavily uses the term "dark knowledge" , but its core mechanism is simply standard (decoupled) logit distillation. So, what is dark knowledge?

**Questions:**

See weaknesses.

---

> ### Author Response · Authors · 2025-11-27
> **Response to Reviewer qzLe**
>
> We thank the reviewer for the detailed summary. We would like to clarify that, while our framework incorporates components inspired by prior work (e.g., decoupled KL decomposition from Zhao et al., 2022), our contributions go substantially beyond directly applying these techniques.
>
> (1). Zhao et al. (2022) study TCKD and NCKD purely in the context of standard single-modal classification, assuming a single unimodal teacher and treating ID classification as the only objective. In contrast, decoupled KD has never been explored in multimodal OOD detection, and extending it to this setting is not straightforward:\
> (i) Near- vs. far-OOD has no analogue in the original DKD:
> The distinction between how TCKD and NCKD influence near- and far-OOD behaviour arises only in the OOD setting.
> Zhao et al. do not study OOD tasks at all—let alone differentiate between near-OOD and far-OOD.
> Our work is the first to reveal, analyze, and empirically validate these OOD-specific behaviors.\
> (ii) Teacher construction is fundamentally different:
> Zhao et al. employ a single unimodal teacher, whereas multimodal OOD requires constructing a holistic teacher distribution from a diverse ensemble of modality-specific models—requiring the uncertainty modeling introduced in our method.\
> (iii) Modality imbalance and underperforming experts:
> In multimodal OOD, some modalities are weaker and highly overconfident. Decoupled KD provides no mechanism for handling such modality asymmetry; our framework introduces likelihood-based teacher weighting to address this issue.\
> (2). To our knowledge, weighted ensemble distillation has not been used to model multimodal uncertainty.
> Our approach is the first to weight unimodal teachers according to modality-specific likelihoods (Eq. 3), explicitly down-weight underperforming modalities, and distill these predictive uncertainties into a joint classifier to improve multimodal OOD detection.
>
> We sincerely thank the reviewer for the feedback and for acknowledging the clarity and readability of the paper.
> Below, we address the concerns raised in the review:
>
> **Q1.** "plug-and-play" framework.\
> **A1.** We thank the reviewer for this thoughtful observation. We would like to clarify that our method is designed to act as a plug-and-play framework not only for existing SOTA multimodal OOD methods, but for any given set of multimodal models, including simple baselines that do not employ any OOD-specific training.
> A key motivation behind our work is that many real-world systems already contain pretrained unimodal/multimodal models trained only on ID data, but retraining these large models with complex SOTA multimodal OOD algorithms is often computationally prohibitive. Our framework addresses this limitation:
> - It does not require retraining the backbone models, unlike A2D, AN, DPU, or FM.
> - It mines intrinsic OOD-relevant signals already present in the unimodal experts.
> - It provides a lightweight, end-stage enhancement of OOD sensitivity by training only a classifier head for a few epochs.
>
> This is fundamentally different from the SOTA methods, which require full end-to-end retraining of multimodal backbones with contrastive objectives or synthetic outlier generation.
> Importantly, our experiments show that this approach is not merely a marginal boost.
> A simple integration of our framework into a vanilla baseline (“Base+Ours”) yields substantial OOD improvements, often matching or surpassing fully retrained SOTA methods (e.g., in Table 3 for near-OOD, Base+Ours outperforms A2D, AN, DPU, and FM in many benchmark cases).
> Thus, our “plug-and-play” claim refers to the fact that the method can be applied to any multimodal model set—even models that were never explicitly trained for OOD detection—and immediately improve their OOD sensitivity without altering the original task performance or requiring costly end-to-end retraining.
> In addition, evaluating our framework on top of existing SOTA multimodal OOD methods is intentional and serves a specific purpose. Since our method is model-agnostic, we aim to demonstrate that it can consistently improve any multimodal model set (including strong SOTA backbones) without modifying their architectures or retraining them end-to-end.
> By keeping the SOTA backbones fixed and applying our framework, we show that it provides additional performance gains of up to 30\%, even on models that already employ complex multimodal training strategies.

---

> ### Author Response · Authors · 2025-11-27
> **Response to Reviewer qzLe**
>
> **Q2.** "poor ID classification performance" equates to "poor OOD detection capability"\
> **A2.** We thank the reviewer for raising this important question. We would like to clarify that our method (Section 3.3) does not assume that “poor ID classification performance = poor OOD detection capability.” In particular, our weighting scheme does not rely on ID accuracy; instead, it uses the likelihood computed on training data, which reflects how well calibrated and confident a modality’s predictions are. Unlike accuracy, which only checks whether the top-1 prediction matches the ground truth, CE loss penalizes miscalibration: even a correct prediction with low confidence (e.g., 51\%) incurs a high loss. Thus, CE captures both correctness and confidence.\
> The reviewer’s example in Fig. 1 (optical flow) actually supports our design. A modality may appear acceptable in terms of ID accuracy yet still be harmful for OOD detection because its predicted probabilities are overconfident. We also cannot rely on OOD data for weighting, since OOD examples are not available during training. Even when synthetic OOD samples are used, their effectiveness depends heavily on the quality of the synthetic data. CE loss offers a lightweight and reliable alternative: it captures modality miscalibration far more effectively than accuracy and therefore used as a suitable signal for weighting the unimodal teachers.\
> Empirically, our weighting improves OOD detection while avoiding penalization of “useful” modalities. As shown in our ablation studies (Table 5), replacing likelihood-based weighting with uniform weighting (“Uniform KD”) consistently reduces performance across all datasets, further validating the effectiveness of our approach.\
> It is also important to emphasize that the weighting is only one component of a broader framework that explicitly incorporates uncertainty-aware logits. Our method uses dark knowledge (logit-level information) to correct overconfidence.
> Even if a modality has high ID accuracy but poor OOD detection (as in the reviewer’s example), this weakness is corrected through: NCKD (which reduces far-OOD overconfidence), and TCKD (which handles ambiguous ID and near-OOD samples).
>
> **Q3.** Hyperparameters $\alpha$ and $\beta$.\
> **A3.** In response, we have added a discussion clarifying this point in Appendix A.3 of the updated manuscript.
> As discussed in Section 3.4 and demonstrated in the hyperparameter analysis in Appendix A.3, our method is robust across a wide range of $\alpha$ and $\beta$ values. While near-OOD typically benefits more from larger $\alpha$ (TCKD) and far-OOD from larger $\beta$ (NCKD), both components individually improve performance.
> For real-world deployment where the type of OOD shift is unknown, we recommend a balanced and task-agnostic configuration such as $\alpha=0.5$ and $\beta=0.5$, which performs robustly across all benchmarks and OOD scenarios.
> Alternatively, practitioners may adopt a proxy-OOD tuning strategy, which is conventional in prior OOD detection (Hendrycks et al., 2019; Zhang et al., 2023; Dong et al., 2024).\
> In addition, many real-world applications naturally prioritize one type of OOD. Some systems primarily focus on near-OOD detection (e.g., fine-grained classification, medical imaging, product or defect inspection) (Zhang et al., 2023), where far-OOD samples are trivial and typically filtered earlier in the pipeline.
> Conversely, some applications prioritize far-OOD (e.g., OCR scanners detecting non-text inputs need to be rejected early in pipeline).
> Our framework allows practitioners to adjust $\alpha$ and $\beta$ according to such priorities, but importantly, no prior knowledge of the exact OOD type is required to benefit from the method, as it consistently improves both near- and far-OOD detection under a wide range of settings.
>
> **Q4.** Usage of the word "Intrinsic".\
> **A4.** We thank the reviewer for this thoughtful observation. We agree that our method introduces a new joint classifier and a distillation loss; however, the term intrinsic in our paper refers to the source of the knowledge being exploited.
> The knowledge used in our framework comes entirely from the given set of pretrained unimodal models (video, flow, audio).
> We do not introduce any external teachers, auxiliary OOD datasets, synthetic outliers, or architectural modifications.
> Thus, by intrinsic OOD detection capability, we refer to the complementary uncertainty knowledge inherently encoded in the unimodal experts, information that prior multimodal OOD methods neglect or discard. Our framework reveals this inherent capability, but does not introduce new external knowledge or retrain modality-specific backbones.
> To avoid confusion, we are happy to clarify this phrasing in the revision.

---

> ### Author Response · Authors · 2025-11-27
> **Response to Reviewer qzLe**
>
> **Q5.** I don't agree that this is a Post-hoc method.\
> **A5.** We thank the reviewer for pointing this out. We would like to clarify that our paper does not claim that the proposed method is a post-hoc OOD scoring method such as MSP or Energy.
> The reason we evaluate compatibility with several post-hoc OOD scoring methods in the ablation study is to show that our framework enhances the reliability of classifier logits, making them more suitable for a variety of existing post-hoc scoring rules.
> We hope this clarifies the misunderstanding.
>
> **Q6.** What is dark knowledge?\
> **A6.** We thank the reviewer for raising this interesting question. In our paper, the term dark knowledge is used in the standard sense introduced by Hinton et al. (2015) (as discussed in line 061 of the manuscript), and widely adopted in subsequent work (e.g., Zhao et al., 2022).
> While the underlying mechanism is logit distillation, we use the term dark knowledge to specifically refer to the uncertainty-aware, modality-dependent information encoded in the unimodal logits.
> This information goes beyond what one-hot labels provide and is what our framework extracts and leverages to improve multimodal OOD detection.
>
> Thank you once again for your thoughtful and constructive feedback.
> We have carefully addressed all your comments and incorporated the recommended revisions accordingly, which we believe have greatly enhanced the quality and clarity of our paper.

---

### Official Review · Reviewer_XoF3 · 2025-10-31

**Soundness:** 3
**Presentation:** 3
**Contribution:** 3
**Rating:** 6
**Confidence:** 4

**Summary:**

The paper proposes a novel framework, self multimodal OOD distillation, to improve out-of-distribution (OOD) detection in multimodal settings. The central problem it addresses is that existing fusion-based methods often fail to account for the varying OOD detection capabilities of different modalities and do not exploit the uncertainty information present in the logits. The core contribution is a method to train a joint classifier. Instead of using one-hot labels, it uses a dynamically generated soft target (dark knowledge) for distillation. The method is presented as a model-agnostic, plug-in module that can be applied on top of existing multimodal OOD methods. The authors demonstrate significant performance gains across five datasets, two tasks, and several state-of-the-art baselines.

**Strengths:**

1. The paper's primary idea is both intuitive and novel. The motivation that different modalities have different intrinsic OOD detection strengths is a key insight.

2. The methodology is well-founded, building logically from a Bayesian-inspired ensemble to a practical and decoupled distillation loss.

3. The paper is well-written and easy to follow.

4. The paper provides extensive experiments, showing the effectiveness and versatility of the proposed method.

**Weaknesses:**

1.  The method needs a priori knowledge of the task (near-OOD vs. far-OOD) to set the α and β hyperparameters (line 409).  In real-world deployment, the type of OOD shift is unknown.

2. The α and β values are fixed values chosen for each task. There is no discussion of how these values were chosen or how sensitive the model's performance is to them.

**Questions:**

1. How would you recommend a practitioner set α and β in a real-world scenario where the type of OOD (near/far) is unknown?

2. A brief sensitivity analysis for the α and β parameters?

---

> ### Author Response · Authors · 2025-11-27
> **Response to Reviewer XoF3**
>
> We sincerely thank the reviewer for the positive and encouraging feedback. We appreciate the recognition of our core idea—that different modalities inherently possess varying strengths in OOD detection—and that exploiting this modality-aware uncertainty through self multimodal OOD distillation is both intuitive and novel. We are glad that the reviewer considers the methodology to be well-founded and appreciates the logical transition from a Bayesian-inspired ensemble formulation to our practical decoupled distillation framework.
> We are also grateful for the positive comments on the clarity of the writing and the comprehensiveness of our experimental validation. Our goal was to design a model-agnostic, plug-and-play framework that can effectively enhance OOD detection performance while preserving ID accuracy, and we are pleased that the reviewer finds the approach effective and versatile across multiple datasets and tasks.
> Below, we address the concerns raised in the review:
>
> **Q1.** How would you recommend a practitioner set $\alpha$ and $\beta$ in a real-world scenario where the type of OOD (near/far) is unknown?
>
> **A1.** We thank the reviewer for this insightful question. As discussed in Section 3.4 and demonstrated in the hyperparameter analysis in Appendix A.3, our method is robust across a wide range of $\alpha$ and $\beta$ values. While near-OOD typically benefits more from larger $\alpha$ (TCKD) and far-OOD from larger $\beta$ (NCKD), both components individually improve performance.
> For real-world deployment where the type of OOD shift is unknown, we recommend a balanced and task-agnostic configuration such as $\alpha=0.5$ and $\beta=0.5$, which performs robustly across all benchmarks and OOD scenarios.
> Alternatively, practitioners may adopt a proxy-OOD tuning strategy, which is conventional in prior OOD detection (Hendrycks et al., 2019; Zhang et al., 2023; Dong et al., 2024).
>
> In addition, many real-world applications naturally prioritize one type of OOD. Some systems primarily focus on near-OOD detection (e.g., fine-grained classification, medical imaging, product or defect inspection) (Zhang et al., 2023), where far-OOD samples are trivial and typically filtered earlier in the pipeline.
> Conversely, some applications prioritize far-OOD (e.g., OCR scanners detecting non-text inputs need to be rejected early in pipeline).
> Our framework allows practitioners to adjust $\alpha$ and $\beta$ according to such priorities, but importantly, no prior knowledge of the exact OOD type is required to benefit from the method, as it consistently improves both near- and far-OOD detection under a wide range of settings.
> We have added a discussion clarifying this point in Appendix~A.3 of the updated manuscript.
>
> **Q2.** A brief sensitivity analysis for the $\alpha$ and $\beta$ parameters?
>
> **A2.** We thank the reviewer for the suggestion. In response, we have added a new section titled “Hyperparameter Analysis” (Appendix A.3), where we provide a dedicated sensitivity study of the $\alpha$ and $\beta$ parameters.
> As discussed in Section 3.4, increasing $\beta$ (which emphasizes the NCKD component) generally benefits far-OOD detection, whereas higher $\alpha$ values tend to be more effective for near-OOD.
> Importantly, across all tested $(\alpha,\beta)$ combinations, the performance consistently remains above the baseline, demonstrating that our framework is stable and robust under a wide range of hyperparameter settings.
>
> We sincerely appreciate your thoughtful and constructive feedback. Your suggestions were very insightful and helped us significantly improve both the clarity and overall quality of the paper. We have carefully addressed all the comments and incorporated the recommended revisions accordingly.

---

### Author Response · Authors · 2025-12-03
**Author Summary Comment**

We would like to note that due to the recent data-leak incident, the system became locked for reviewers very shortly after we submitted our rebuttal. As a result, reviewers did not have the opportunity to respond to our clarifications.

Our rebuttal addresses all concerns raised in the reviews.
We hope these clarifications are helpful for the AC during the meta-review process, and we sincerely appreciate your time and consideration.

---

### Meta-Review · Area_Chair_EvwF · 2026-01-07

**Summary:**

This paper proposes a self multimodal OOD distillation framework that leverages dark knowledge (logit-level uncertainty) from unimodal experts to improve multimodal out-of-distribution detection. By weighting modalities and distilling both target and non-target class knowledge into a joint classifier, it aims to mitigate underperforming or overconfident modalities. Extensive experiments across multiple datasets show consistent improvements over existing multimodal OOD baselines.

**Reviewer Concerns:**

Reviewers’ concerns center on (1) limited novelty, arguing the method largely adapts existing decoupled knowledge distillation and ensemble distillation techniques rather than introducing fundamentally new ideas; (2) questions about the experimental fairness and interpretation of gains, since improvements may partly stem from additional classifier-head fine-tuning rather than true advances in multimodal OOD modeling; and (3) practical ambiguity in hyperparameter selection and conceptual clarity (e.g., the meaning of “intrinsic,” “dark knowledge,” and whether the method is post-hoc). Among these, the novelty concern and the critique that gains may reflect extra fine-tuning rather than intrinsic multimodal OOD advantages are not fully resolved to all reviewers’ satisfaction.

**Reviewer Scores:**

6,2,4

---

### Decision · Program_Chairs · 2026-01-26

Reject